# PET-CT in Clinical Adult Oncology: I. Hematologic Malignancies

**DOI:** 10.3390/cancers14235941

**Published:** 2022-11-30

**Authors:** Ahmed Ebada Salem, Harsh R. Shah, Matthew F. Covington, Bhasker R. Koppula, Gabriel C. Fine, Richard H. Wiggins, John M. Hoffman, Kathryn A. Morton

**Affiliations:** 1Department of Radiology and Imaging Sciences, University of Utah, Salt Lake City, UT 84132, USA; 2Department of Radiodiagnosis and Intervention, Faculty of Medicine, Alexandria University, Alexandria 21526, Egypt; 3Department of Medicine, Division of Hematology, Huntsman Cancer Institute, University of Utah, Salt Lake City, UT 84132, USA; 4Intermountain Healthcare Hospitals, Murray, UT 84123, USA

**Keywords:** imaging, FDG, positron emission tomography, PET, lymphoma, leukemia, multiple myeloma, B-cell lymphoma, T-cell lymphoma

## Abstract

**Simple Summary:**

Positron emission tomography (PET); typically combined with computed tomography (CT), has become a critical advanced imaging technique in oncology. With PET-CT; a radioactive molecule (radiotracer) is injected in the bloodstream and localizes to sites of tumor because of specific cellular features of the tumor that accumulate the targeting radiotracer. The CT scan; performed at the same time; facilitates better visualization of radioactivity from deep or dense structures; and to provide detailed anatomic information. PET-CT has a variety of applications in oncology, including staging, therapeutic response assessment, restaging, and evaluation of suspected recurrence. This series of six articles provides an overview of the value, applications, and imaging and interpretive strategies of PET-CT in the more common adult malignancies. The current article addresses the use of PET-CT in hematologic malignancies.

**Abstract:**

PET-CT is an advanced imaging modality with many oncologic applications, including staging, assessment of response to therapy, restaging and evaluation of suspected recurrence. The goal of this 6-part series of review articles is to provide practical information to providers and imaging professionals regarding the best use of PET-CT for the more common adult malignancies. In the first article of this series, hematologic malignancies are addressed. The classification of these malignancies will be outlined, with the disclaimer that the classification of lymphomas is constantly evolving. Critical applications, potential pitfalls, and nuances of PET-CT imaging in hematologic malignancies and imaging features of the major categories of these tumors are addressed. Issues of clinical importance that must be reported by the imaging professionals are outlined. The focus of this article is on [^18^F] fluorodeoxyglucose (FDG), rather that research tracers or those requiring a local cyclotron. This information will serve as a resource for the appropriate role and limitations of PET-CT in the clinical management of patients with hematological malignancy for health care professionals caring for adult patients with hematologic malignancies. It also serves as a practical guide for imaging providers, including radiologists, nuclear medicine physicians and their trainees.

## 1. Introduction

PET-CT is an invaluable advanced diagnostic imaging modality in oncology with a variety of applications, including initial staging of cancer, assessment of response to therapy, restaging, and assessment of suspected recurrence. This is particularly true for lymphomas and other hematological malignancies, which in typical practice comprise most referrals for positron emission tomography combined with computed tomography (PET-CT) utilizing [^18^F] fluorodeoxyglucose (FDG). While a variety of newer PET radiopharmaceuticals have recently been introduced, FDG is the radiopharmaceutical that is used almost exclusively for imaging hematological malignancies. Because PET-magnetic resonance imaging (PET-MR) scanners are in limited use in the U.S., the focus herein is on PET-CT because if its widespread availability. However, basic principles described in this review are also applicable to PET-MR.

Many categories and subtypes of hematologic malignancies exist, many rarely seen. This review addresses the most common categories of hematologic malignancies encountered in adult patients in whom FDG PET-CT plays an important role. This article also focuses on FDG PET-CT, rather than research tracers or those that require a local cyclotron. There is a large body of literature that relates the tumor characteristics on FDG PET-CT to prognostic outcome, which in some cases may alter management. Currently, clinical decision-making and patient care algorithms are typically based on the histologic, molecular, and imaging features of these tumors, and not on FDG PET-CT-derived prognostic information. For this reason, this report places greater emphasis on the role of FDG PET-CT in initial staging, as well as on both interim and end-of-treatment therapeutic assessment. It also discusses the important clinical questions that should be specifically addressed and reported with FDG PET-CT. The typical PET-CT characteristics of the most common categories of hematologic malignancies are also addressed. The targeted readers for this review are imaging providers, including radiologists, nuclear medicine physicians, and their trainees. The information is also important to medical and surgical professionals treating adult cancer patients.

## 2. Hematologic Malignancies

Hematologic malignancies (HM) constitute a heterogeneous group of disorders with diverse incidences, prognoses, and etiologies. These disorders arise from either blood forming cells (i.e., bone marrow) or immune response-related cells. They can be broadly classified as mature B-cell neoplasms (including B-cell lymphomas and plasma cell dyscrasias), mature T-cell/natural killer cell lymphomas, Hodgkin lymphoma (HL), post-transplantation lymphoproliferative disorders (PTLD), and dendritic neoplasms. The most recent classification for hematologic malignancies reflects the increase in our understanding of these disorders, and includes new subtypes recently implemented due to the advances in the field of hematology/oncology [1,2]. Increasingly, molecular and genomic markers are employed in decisions regarding the use of targeted therapies, and in eligibility for clinical trials. These molecular features are likely to further change classifications of lymphomas and other hematological malignancies in the future.

### 2.1. Classification of Lymphomas

Malignant lymphomas are considered the most common adult hematologic malignancies, arising from uncontrolled clonal proliferation of lymphocytes or their precursors. They encompass more than 50 unique subtypes that differ significantly in histology, etiology, clinical behavior, and clinical management. They are most broadly classified by the 2016 WHO classification as non-Hodgkin lymphoma (NHL, 90%), and Hodgkin lymphoma (HL, 10%) [1]. HL is primarily B-cell in origin and can be further divided into classical and nonclassical subtypes. Similarly, NHL can be subclassified as either B-cell lymphoma (BCL), T-cell lymphoma (TCL) or NK-cell neoplasms, based on their cellular receptors (Table 1) [1]. Since the 2016 WHO classification, precursor leukemias/lymphomas have generally been considered as a separate category [2]. The WHO classification is scheduled for further revision in 2022.

Lymphomas can also be classified as grade of aggressiveness, either indolent (low-grade) or aggressive (high-grade) types. Most low-grade lymphomas are NHL in origin, in addition to the nodular lymphocyte predominant subtype of HL. Indolent lymphomas are less aggressive but are more difficult to cure. Given their lower tumor proliferation, they respond less well to conventional chemotherapy regimens [3]. Lymphoma can involve almost any organ. However, at onset, HL usually presents with nodal disease, with or without splenic involvement. Extra-nodal disease is more often encountered in NHL [1]. It is beyond the scope of this article to extensively review each subtype of lymphoma. Rather the lymphomas most encountered in clinical practice will be succinctly addressed.

### 2.2. Imaging Strategies for Lymphoma

Various imaging modalities have been implemented for lymphoma, including ultrasound, CT, MRI, and PET-CT. Ultrasound can be used for assessment of superficial structures and to facilitate biopsy. Historically, CT imaging was considered the cornerstone for lymphoma imaging. Nowadays, CT is merged with FDG PET to provide both anatomic and functional information. CT of the neck, chest, abdomen, and pelvis can be used as a stand-alone modality in imaging of low-grade lymphomas, or those that are poorly FDG-avid [4]. MRI is usually typically reserved for assessment of the neural axis, pelvic structures, and characterization of musculoskeletal involvement [5]. However, in initial staging of lymphomas, MRI has been shown to be slightly superior to whole body CT, and similar in performance to FDG PET-CT, with the added achievement in avoidance of radiation exposure [6]. As reviewed by Albano et al., whole-body MRI with diffusion weighted imaging (DWI) not only shows promise in staging, but also in assessing response to treatment [7]. As such, future attempts to standardize MRI sequences and metrics of response are needed and may change the landscape of future options for imaging lymphoma.

FDG PET-CT is currently the preferred modality for staging and response assessment of all high-grade lymphomas or those that are FDG avid [8]. FDG avidity is an indicator of enhanced glycolytic metabolism characteristic of many lymphoma cells. In the case of Hodgkin lymphoma (HL), pleomorphic CCR-4 expressing inflammatory cells recruited by the Hodgkin and Reed Sternberg cells comprise most of the tumor mass and are also highly metabolically active. An awareness of many pitfalls associated with PET-CT imaging is essential in accurate image interpretation, including inflammatory changes, thymic rebound following treatment, pharmacologic and stress-mediated hematopoiesis (marrow stimulation) and post-surgical/radiation inflammatory changes, all of which may result in false positive results. Immunotherapies may also result in flare phenomena and pseudo progression, which can complicate assessment of treatment response.

#### 2.2.1. FDG PET-CT Methodology

The CT component of FDG PET-CT can be performed with or without contrast and with a conventional or low dose CT. Although contrast enhanced CT, when performed with PET, reveals more findings and greater tissue characterization compared to non-contrast CT studies, the difference between the two techniques has been reported to rarely change overall patient management and the plan of treatment [9]. However, if a low dose, non-contrast enhanced CT is performed with PET, it may be necessary to perform a separately acquired contrast enhanced CT scan to confirm the anatomic nature of foci of hypermetabolism and to accurately measure the size of lesions. Ideally, both should be acquired close to the same time, sequentially if possible. However, in the United States (US), insurance reimbursement may be complicated if both studies are performed on the same day. Concerns that iodinated contrast media could alter measured metabolic activity in areas of interest due to attenuation artifact are unfounded [10,11]. Similarly, performing a PET-CT scan with oral contrast media is very helpful in assessing intraabdominal structures, particularly in pediatric patients or those with a low body mass, in whom a lack of intraabdominal fat to separate loops of bowel create interpretive challenges.

Considerations of dosimetry often drive protocol selection for FDG PET-CT. A wide range of FDG PET-CT protocols exist, with variables being the dose of FDG employed, the extent of the body scanned, and the purpose of the CT portion of the scan [Marti-Climent]. Equipment capability primarily drives the dose of FDG utilized. The dose of FDG administered for a PET scan ranges widely between 240–433 MBq [12]. Newer time-of-flight and digital PET-CT scanners allow for a dose at the lower range of FDG to be administered, or shorter imaging times [13,14]. For the PET portion of the exam, an average dose of 10 mCi (370 MBq) of FDG delivers an effective dose of approximately 6.66 mSv (+/−0.74) to adult males and 8.14 mSv (+/−0.74) to adult females [15]. The body range scanned and the purpose of the CT scan (attenuation correction, lesion localization or CT diagnosis) affects the choice of CT protocol employed. The effective dose from the typical eyes-through-thighs CT portion for a moderate low dose CT, intended to facilitate lesion localization (120 kEv, 40–80 mA, ~39 mAs) is 5.3 mSv (+/−1.0) for adult males and 4.6 mSv (+/−0.8) for adult females [15]. For an eyes-through-thighs (torso) scan, conventional diagnostic CT without dose modulation delivers up to 3 times the radiation exposure as does a low dose CT. When this type of CT is used (120 kEV, 223–236 mA, ~120 mAs), an adult male receives an effective dose of 17.4 mSv (+/−2.7), and an adult female receives 13.4 mSv (+/−5.5). Total radiation exposure is higher for CT portions of the exam that image from the top of the head through the feet. Patient-specific CT exposure parameters and the inclusion of CT dose modulation can result in a significant reduction in the radiation dose from the CT portion of the exam, without a significant sacrifice of diagnostic quality [16,17].

For many types of tumors, the National Comprehensive Cancer Network (NCCN) states that a separate CT is not necessary if a PET-CT can be performed with an adequate diagnostic quality of CT. Therefore, if the use of a low dose CT requires additional conventional CT scan for diagnostic certainty, the dosimetric advantage gained by performing a low dose CT with the PET scan is essentially lost. In pediatric patients, however, a moderate CT dose reduction is appropriate and typically employed. The use of oral, and in some case IV contrast, even with a lower dose CT, improves tissue definition and contrast. Although PET-MRI scanners are not widely available and adoption in standard practice has been relatively slow, the advantages gained by this technology are significant in terms of dose reduction, without sacrificing the quantitative advantages of PET.

The appropriate axial range of FDG PET-CT for lymphoma has not been established for different types of lymphoma but, in general, a torso scan (“eyes-to-thighs”) is adequate for most types of lymphoma. A whole-body scan (“top-of-head-through-feet”) is typically employed for lymphomas that are prone to distal or cutaneous involvement, as are many T-cell lymphomas, or in cases where focal bone lesions are known [18]. In multiple myeloma, the region scanned should extend from the skull vertex through at least the knees. Where brain or CNS involvement is known or suspected, an additional, separately acquired FDG PET-CT of the brain should be performed, because whole body technique is not adequate for brain assessment. Even so, FDG PET-CT detection of lymphoma involvement of the brain involvement may be limited by the normal high background of activity in the brain, and a heavy reliance upon MRI imaging is usually necessary.

An important component in optimizing the methodology of FDG PET-CT is the assurance that professionals with the appropriate expertise be available to interpret the exam. Imagers who are well-trained in CT are critical to the correct interpretation of the studies. In the US, radiologist are, as a requirement of their training and certification by the American Board of Radiology (ABR), able to achieve responsible user status for the receipt of radiopharmaceuticals used in nuclear medicine. They are also trained in the interpretation of nuclear medicine studies, including PET-CT, with the requirement of 700 h of dedicated nuclear clinical medicine training plus 80 h of additional focused didactic training. In the US, both radiologists and non-radiologist may be certified by the American Board of Nuclear Medicine (ABNM), which requires non-radiologists to undergo 6 months of training in CT. Non-radiologist ABNM diplomats also frequently elect to do additional fellowships in PET-CT. In the Netherlands, nuclear medicine training has also been integrated into diagnostic radiology. In most other countries, radiologists are not certified to receive radiopharmaceuticals nor to report the findings of PET-CT. A collaborative interpretative approach between radiology and nuclear medicine practitioners is necessary in the countries that do not provide dual radiology/nuclear medicine training. This integrative approach may create complexity and some financial barriers, but it is necessary from a regulatory perspective in these countries and may also offer optimized and coordinated expertise, even in countries where dual training is employed.

#### 2.2.2. Analysis of Metabolic Activity on FDG PET-CT

Attenuation-corrected PET images can be analyzed by a myriad of different semi-quantitative parameters that provide prognostic information, to allow comparison to be made that predict treatment response for a specific patient, and to allow imaging harmonization for clinical trials. The most used of these is the standardized uptake value, or SUV. For lymphomas, a ratio between the SUV in the lesion of interest to that of a normal reference background tissue, such as mediastinal blood pool or liver, is commonly employed. These standardized uptake values are typically normalized to some index of patient size, such as body weight, lean body mass, or body surface area. The standardized uptake value (SUV) of the lesion may be represented as a SUV of the “hottest” (most metabolically active) pixel in a region of interest (SUVmax), an average SUV (SUVavg), or a peak SUV (SUVpeak), which is the average activity in a 1 cm^3^ region of tumor around the “hottest” pixel. SUVpeak has been shown to be more stable than other parameters in a large region of relatively homogeneous metabolic activity (such as the liver) but is less widely used than is SUVmax [19]. The stability and reproducibility of these parameters varies as a function of the size and heterogeneity of the target lesion, the interval from uptake to imaging as well as patient preparation, acquisition, processing and reconstruction protocols and individual scanner performance [20]. In clinical practice, every attempt should be made to faithfully reproduce these factors from one scan to the next, and to utilize the same equipment and protocol for each imaging session.

In addition to typical SUV measurements described above, a variety of newer semiquantitative parameters have been proposed that may have prognostic value, such as total lesion glycolysis (TLG) and total metabolic tumor volume (TMTV). The most adopted of these measurements is TMTV. This is an index of overall disease burden and is derived from thresholding and segmentation applications [21,22]. TMTV can be combined with an index of dissemination of disease, derived from a calculation of the furthest distance between two sites of disease (SDmax). These parameters are normalized to patient size. High TMTV is an independent predictor of less favorable survival [23]. Characterization of textural features such as metabolic and intravenous iodinated contrast enhancement heterogeneity have been derived using radiomic measurements of voxel SUV histograms or SUV spatial organization in a tumor volume. The field of radiomics is burgeoning field that shows great promise in refining the biological characterization of tumors by imaging [24]. High heterogeneity in metabolic activity may indicate intratumoral biological variability, genetic instability and may predict a less favorable outcome [25,26].

#### 2.2.3. Baseline FDG PET-CT for Staging Lymphoma

The initial step in defining the characteristics of lymphoma is the establishment of stage and grade. The accurate initial staging of lymphoma is critical in establishing the extent of disease, predicting 5-year survival rates, and in determination of appropriate treatment options. The earliest system implemented for staging lymphoma was the Ann Arbor system, which was originally developed for HL and later extended to NHLs [27]. In 2014, a new classification system was published as the outcome of an international conference held in Lugano, Switzerland, that has been adopted in most clinical practices and is applicable to both HL and NHL (Table 2) [8]. In HL and NHL, Lugano staging is classified by Roman numerals I-IV defining extent of disease. In addition to the numerical staging classification, HL are further classified by letters designating factors of importance: A (no symptoms), B (symptoms of either unexplained weight loss, drenching night sweats, or fever), X (bulky disease), E (extracapsular nodal spread of stage I or II), and S (presence of splenic involvement).

NHLs share a similar staging system to that of HL, with additional histologically features defining the grade of disease, which predominantly features the cell division pattern defined as low-grade (meaning indolent), intermediate-grade, or high-grade (aggressive). Genomic features of NHL, particularly aggressive DLBCL, primary mediastinal B-cell lymphoma, and some follicular lymphomas provide important prognostic information and based on mutational patterns, predict susceptibility to targeted therapies.

The Lugano conference established FDG PET-CT as a critical modality in guiding management of all FDG-avid lymphomas, which includes HL and DLBCL, except for lymphomas that are typically low in metabolic activity, such as small lymphocytic lymphoma, marginal zone lymphoma and some cutaneous lymphomas [8,28]. A baseline FDG PET-CT should be performed before initiation of any treatment, as any therapeutic intervention may rapidly alter tumor metabolism, even within only a few days, thus affecting scan sensitivity (Figure 1) [28]. In both HL and DLBCL, inclusion of the prognostic index of total metabolic tumor (TMTV) has been shown to lead to a change in patient management and treatment [29]. TMTV has been reported to better define risk than the Ann Arbor Staging [28]. However, the use of TMTV as defined by FDG PET-CT has not yet been widely adopted in clinical practice.

FDG PET-CT is more accurate than CT in staging of aggressive lymphomas, showing increased sensitivity and specifically for detecting extra-nodal sites of disease. Compared to CT, it is reported that FDG PET-CT upstages 10–30% of patients with lymphoma, primarily in the detection of additional sites of disease [28]. It is also critical in excluding additional sites in patients thought to have limited disease who may be eligible for management by radiation therapy alone. However, FDG PET-CT is not routinely required for all indolent lymphomas, where CT alone can still be used. In interpreting FDG PET-CT scans in lymphoma, it is important to specifically document the presence of higher risk variables that are also included in the Lugano staging system, such as the presence of nodal extracapsular extension, bulky disease (a single nodal site or nodal conglomerate > 10 in any diameter), or extra-nodal sites of disease. Nodal sites include lymph nodes, thymus and the pharyngeal lymphatic (Waldeyer’s) ring of tonsillar tissue. The spleen is considered a nodal site for HL but is extra-nodal for NHL [30]. FDG PET is most limited in staging in lymphomas that tend to be lower in metabolic activity, or those that occur with a propensity for regions that are typically high in metabolic activity.

As a rule, more aggressive lymphomas tend to be more metabolically active than indolent lymphomas [31]. Marginal zone lymphomas, small-cell lymphocytic lymphoma and many cutaneous or enteropathy-associated T-cell lymphomas (EATL) may be relatively low in metabolic activity, although FDG PET-CT may nonetheless outperform CT in some of these cases [32]. For example, in refractory celiac disease, FDG PET-CT may be helpful in directing biopsy for EATL [33]. With a few exceptions, an FDG PET-CT scan can aid in subclassifying some lymphomas into aggressive or indolent subtypes based on lesion’s standardized uptake values (SUVs) and intratumoral metabolic heterogeneity. For example, B-cell lymphomas with tumor SUVmax > 10 at baseline and high tumor heterogeneity often associated with a more aggressive course [3]. Exceptions to the trend toward higher grade lymphomas being higher in FDG activity do exist. For example, lower grade follicular lymphomas (FL) can be high in metabolic activity although SUVmax > 10 is typically associated with Grade III or transformed FL [34]. FDG PET-CT may also suggest whether different tumor subtypes or both untransformed and transformed lesions may coexist in the same patient, manifesting as tumor sites with different metabolic signatures. This may support the need for tissue sampling of different lesions, guided by FDG PET-CT.

In staging of lymphoma, there may be additional value in newer semiquantitative features derived from FDG PET-CT. These include total lesion glycolysis (TLG), total metabolic tumor volume (TMTV), degree of disease dissemination (SDmax), and measurements of heterogeneity in metabolic activity within the tumor. TMTV has been reported to better define risk than the Ann Arbor Staging [28]. Inclusion of the prognostic index of total metabolic tumor (TMTV), has led to change in patient’s management and treatment in both HL and DLBCL [29,35]. In analysis of the large REMARC trial in elderly patient with DLBCL, Cottereau et al. determined that SDmax was highly predictive of a poorer prognosis, independent of disease burden as defined by TMTV [16]. In analysis of this same trial, Vercellino et al. reported that high TMTV at baseline predicts survival, independent of response to therapy [36]. In the recently reported results of the GOYA trial, high baseline TLG and TMTV were both independent predictors of unfavorable prognosis in treatment-naïve patients with DLBCL who were subsequently treated with chemoimmunotherapy [23]. High TMTV has also been shown to predict reduced progression free survival (PFS) and overall survival (OS) in follicular lymphoma [36]. A high degree of heterogeneity of metabolic activity within a tumor mass, or between different sites of tumor on the same baseline exam is indicative of biological, and theoretically clonal, variation and is more likely to predict chemotherapy refractory disease [37]. For example, in DLBCL, metabolic heterogeneity on baseline FDG PET-CT has been reported to result in a worse prognosis [38]. Metabolic heterogeneity has also been shown to predict a poorer prognosis in primary mediastinal B-cell and mantle cell lymphomas [25,26]. These semi-quantitative indices of disease burden, dissemination and metabolic heterogeneity show great promise in prognostic stratification of patients but have not yet reached widespread clinical use and the methodology for measuring these parameters has not yet been formally standardized for clinical lymphoma trials. Whether modification of therapeutic approaches based on these FDG PET indices will improve outcome is an area of needed research.

Baseline FDG PET-CT eliminates the need for bone marrow biopsy (BMB) in many cases of both HL and DLBCL when focal hypermetabolic osseous lesions, with or without intense diffuse marrow involvement, are present. Because of the superior sensitivity of FDG PET-CT, BMB are rarely done with HL. The typical pattern of bone involvement in HL is focal hypermetabolic lesions without a corresponding CT abnormality, which is present in 15% of newly diagnosed HL [39]. Numerous studies have shown superior sensitivity of FDG PET-CT over BMB in HL and DLBCL [39,40,41]. For example, in analysis of the PETAL and OPTIMA >60 multicenter trials diagnosing bone marrow involvement of NHL, it was reported that BMB and FDG PET-CT, respectively, showed sensitivities of 38% (95% CI: 32–45%) and 84% (95% CI: 78–88%), specificities of 100% (95% CI: 99–100%) and 100% (95% CI: 99–100%), positive predictive values of 100% (95% CI: 99–100%) and 100% (95% CI: 99–100%), and negative predictive values of 84% (95% CI: 81–86%) and 95% (CI 93–97%). However, in this trial, because of other adverse clinical factors that drove therapy, there was no significant difference in management of patients as a consequence of greater sensitivity of FDG PET-CT in detecting bone marrow involvement [41]. In the era of FDG PET-CT, BMB is performed in NHL only if the results will change the stage from early to advanced stage disease. There has been considerable work in development of artificial intelligence-radiomic analysis of textural and other features of FDG PET-CT scans in predicting bone marrow involvement, which show promising results [42,43,44,45].

Reliance upon FDG PET-CT as a substitute for bone marrow biopsy is an imperfect process. Definitive identification of marrow involvement is challenging when there is diffuse marrow infiltration, as may be seen with DLBCL, follicular lymphoma and mantle cell lymphoma [29], as well as those patients with a preexisting myelodysplastic disorder, which may result in increased metabolic activity with marrow expansion into the long bones. In such cases, bone marrow biopsy may be required for accurate bone marrow staging [46]. There are a many false positive causes of both focal and diffusely increased metabolic activity within the red marrow that represent potential pitfalls of FDG PET-CT scan. Focal activity can be seen in older patients or smokers, in whom fatty marrow may have replaced red marrow in a heterogeneous pattern, or in marrow reconversion, where red marrow repopulates areas of yellow marrow. Stress-induced hematopoiesis or granulopoiesis may increase diffuse metabolic activity in the marrow. This may result from intercurrent or recent infection, bleeding, or anemia. A paraneoplastic leukemoid reaction characterized by stimulation of granulocyte colony stimulating factor (GCSF) production by the tumor may result in diffusely increased metabolic activity in the red marrow [47]. This is often recognizable because of significantly elevated neutrophil counts in the blood. A final limitation of reliance upon FDG PET-CT for staging the bone marrow is that there is no consistent consensus on what constitutes the upper limit of normal metabolic activity in bone marrow, and this may vary with age and gender. For example, normal young women often have diffusedly increased metabolic activity in the marrow, likely due to menstrual blood loss and chronic iron deficiency anemia. In cases where equivocal marrow involvement may be suggested by FDG PET-CT, the scan may nonetheless aid in directing targeted biopsy to the most metabolically active region.

#### 2.2.4. Interim FDG PET-CT (iPET) for Assessing Early Response to Treatment

Interim FDG PET-CT (iPET) can identify early non-responders so that therapeutic strategy can be reassessed early in the course of treatment. iPET is usually performed following 2 to 4 cycles of chemotherapy with the goal of identifying which patients are more likely to achieve complete response versus patients who will require the addition of consolidative therapy or a change in chemotherapy [28]. iPET can play a critical role in management of HL [28,48], although the role of iPET in management of NHL is less well supported. iPET can detect early treatment response, with a decreased in metabolic activity before any structural changes in tumor volume have occurred. It can also suggest whether histologic transformation has occurred during the course of disease, by detecting change in metabolic signature or newly appeared lesions despite adequate received treatment. This allows for repeat targeted biopsy and therapeutic refinement to reduce toxicity in lower risk disease and allow escalation in patients with a suboptimal response.

The Deauville classification of FDG PET response, first published in 2009, is 5-point PET scoring system based on maximum standardized uptake volume (SUVmax) relative to reference tissues (mediastinal blood pool and liver), that predict the response of lymphomas to treatment (Table 3) [4,49]. Although there is some evidence of age-related changes in metabolic activity in background tissues [50], metabolic activity in the liver and mediastinal blood pool are regarded as being relatively stable from one imaging session to the next. This system was further refined based on the outcome of an international workshop in Menton, FR in 2014 to include a wider range of lymphomas, to apply the scoring system to end-of-treatment FDG PET-CT (ePET) as well as interim PET-CT (iPET), and to introduce the concept of ΔSUVmax from baseline to iPET [51]. The Deauville criteria were then further modified at the workshop in Lugano and published in 2014 [4]. These modifications recommended the use of FDG PET-CT as a gold standard for initial staging all FDG-avid lymphomas that can present with nodal involvement, which included all except for lymphomas typically low in metabolic activity, namely chronic lymphocytic leukemia/small lymphocytic lymphoma, mycosis fungoides, marginal zone lymphoma (MALT), and lymphoplasmacytic lymphoma/Waldenström macroglobulinemia. The conference also introduced the concept of partial metabolic response (PR) for iPET (Table 3). Thus, even with an iPET Deauville score of 4 or 5, a noticeable reduction of metabolic activity would be indicative of chemo-sensitive disease.

There are several technical factors that contribute to uncertainty when utilizing the Deauville scoring system. The initial development of the Deauville classification system was based on visual assessments, which remains the current standard [51]. With the application of semi-quantitative assessments of SUV, several sources of ambiguity arose. First, metabolic activity in tumors tend to increase over time up to 70 min, while activity in the blood pool decreases. The interval from injection to imaging must be rigidly standardized to avoid sources of error. Although greater tumor background activity may be achieved at 70 min post injection, most clinical trials have used an uptake interval of 60 min. The second issue is whether SUVmean, SUVmax or SUVpeak should be utilized for measurements of activity in background reference tissues and in tumors. Partial volume effects result in decrease in measurements of metabolic activity for small lesions. This might be expected to be greater for SUVmean or SUVpeak measurements. Heterogeneity in tumors would also result in lower averaged SUV determinations. Therefore, SUVmax might be considered a better reflection of residual incompletely treated disease, even if it includes only a small portion of the residual tumor mass. Finally, whether SUVmean or SUVmax is used for the background tissues is an important issue, and one that is only rarely addressed [52]. For the liver, a volume of 3 cm^3^ is generally recommended for measurements of metabolic activity. With newer digital scanners and for heavy patients, image noise is more apparent. Therefore, SUVmean may represent a better estimate of activity in blood pool and especially in liver, than SUVmax. However, it must be emphasized that the initial Deauville scoring system was based on visual assessments. Visually, assessments of relative metabolic activity are appreciated as SUVmax (mean intensity cannot be appreciated visually). In addition, multicenter trials tend to use SUVmax of background tissues. Even if visual assessment is used for analysis of treatment response, the reconstruction algorithms and the type of scanner used may change the point spread function and affect the visually appreciable activity in lesions [52]. Therefore, unless a global harmonization consensus is reached, it may be safer to utilize SUVmax rather than SUVmean for both background tissues and tumor. It is also critical to ensure that a given patient is imaged, over time, with the same scanner using the same interval from injection to imaging, as well as application of identical processing and reconstruction algorithms. If standardization cannot be accomplished, the Lugano consensus was that visual assessment of treatment response by PET should take precedence. Visual assessment should be applied to all semiquantitative determinations as an added mechanism of quality assurance.

The benefit of iPET has been most significantly shown with HL, where the results of several published clinical trials have shown that a negative iPET can justify reduction to a less toxic therapeutic regimen without compromising outcome [53,54,55,56,57]. Conversely, escalating to a more aggressive therapy in the face of a positive interim scan in high-risk patients with HD has been shown to have a positive therapeutic effect [57]. In advanced Hodgkin lymphoma, most common PET-adapted strategy is performing PET after 2 cycles of doxorubicin, bleomycin, vinblastine, and dacarbazine (ABVD). If PET has Deauville score of 1–3, then bleomycin, which is more highly toxic, is dropped for the next 4 cycles with no significant difference in PFS at 3 years. About 20% will be persistently PET-positive and typically those patients would be switched to escBEACOPP [58]. However, this strategy of interim PET is being used less frequently now due to approval of brentuximab vedotin, a CD30-directed antibody-drug conjugate (A + AVD), in advanced HL, based on the ECHELON 1 trial. This showed improved efficacy of A + AVD over ABVD, without the toxicity of bleomycin [59]. Recently, a 5-year follow-up of the ECHELON 1 trial confirmed a sustained advantage of A + AVD over ABVD for advanced stage HL [60].

For NHL, a positive iPET scan is predictive of a worse OS and may have value in prognostic risk stratification [61,62,63]. However, although iPET has prognostic value, the benefit of treatment modification as a result of iPET has not been proven for DLBCL and other aggressive NHLs [63,64]. A high biopsy negative rate has been reported in positive iPET in advanced stage DLBCL, with a similar outcome to patients with PET-negative scans [65,66]. Notably, results of the PETAL trial showed no benefit to therapeutic escalation in patients with aggressive DLBCL and T-cell lymphoma with positive iPET scans [66]. However, further randomized studies are warranted.

#### 2.2.5. Immunotherapy and Pseudo Progression/Flare Phenomenon

With the development of new therapies for certain lymphomas, including immune check inhibitors (PD1 inhibitors), there was an urgent need to develop a new response criterion to account for transient tumor pseudo progression, or flare phenomenon, whereby lesions can increase in apparent size and/or metabolic activity due to an influx of inflammatory cells with immunotherapy or CAR-T therapy. In addition, new lesions can appear due to a sarcoid-like inflammatory response within lymph nodes and many organs, even those not originally involved with tumor. All of these can give the false appearance of progression of disease. For this reason, lymphoma response to immunomodulatory response criteria (LYRIC criteria) was developed as an adaptation of the Lugano criteria, with the addition of a category of indeterminate response (IP), where progression is suggested by the imaging findings in presence of clinical stability or improvement [67]. There are three categories of IR:IR(1): ≥50% increase in overall tumor burden based on the summed products of perpendicular diameters (SPD) of up to 6 target lesions in the first 12 weeks of therapy and with clinical stability.IR(2): New lesions or ≥50% increase of previous lesions without a ≥50% increase of overall tumor burden.IR(3): Increased FDG uptake in any lesion without an increase in size or number.

IR is a transient and not a progressive phenomenon. Recommendations for suspected IR, in the face of clinical stability or improvement, is for treatment to continue, but for a repeat biopsy or FDG PET-CT scan to be performed in 12 weeks to determine true progression vs. pseudo progression or flare. Although LYRIC criteria were published in 2016, there has been slow uptake of this in clinical practice. Currently LYRIC criteria are being mainly utilized in clinical trials as secondary end points with immunomodulatory agents such as anti-PD1. There is an increased need for education as to the value of this modification of the Lugano response criteria.

#### 2.2.6. End-of-Treatment FDG PET-CT (ePET) to Confirm Resolution of Disease

End-of-treatment FDG PET-CT (ePET) is typically performed 8–12 weeks after completion of chemotherapy or radiation therapy to avoid post-treatment inflammatory changes, with the longest of these delays advisable following radiation [68]. In patients whose tumors require induction, consolidation and/or maintenance phases of therapy, the ePET is performed following the induction phase. Approximately 50% of aggressive lymphomas such HL and DLBCL may form residual fibrotic/necrotic masses even with successful treatment. By applying the Deauville criteria (Table 3, above), detection of uptake higher than liver background in these masses (Deauville 4 or 5) is consistent with residual active disease, requiring additional therapy to achieve remission. Before initiation of salvage treatment, a biopsy is typically recommended to confirm presence of residual disease. This is important because of the substantial false positive rate of FDG PET-CT. A published meta-analysis reported that ePET results in a false positive rate of 23.1% (95% CI: 4.7–64.5%) for HL and 31.5% (95% CI: 3.9–83.9%) for NHL [69]. Under these circumstances, the FDG PET-CT scan can nonetheless aid in identifying the most accessible suspicious lesion for biopsy. A residual mass is particularly common in primary mediastinal B-cell lymphoma (PMBL). Frequently, serial FDG PET-CT scans may be done before committing to radiation.

A negative ePET, even in patients with advanced or bulky HL or DLBCL, can obviate the need for consolidative radiotherapy. The results of the HD0607 trial showed that a complete metabolic response to treatment as shown by ePET, even when a residual mass is present, allowed for an elimination of consolidative radiotherapy without a decrease in PFS [70]. Similar findings were reported from a large retrospective Canadian study of a cohort of patients treated with six cycles of RCHOP for advanced stage DLBCL [71]. However, for patients who have DS of 4/5 in absence of new areas of disease, radiation may still be of benefit. For follicular lymphoma (FL), FDG PET-CT shows a post-induction advantage over CT alone in predicting outcome [72]. However, elimination of post-induction maintenance based on a negative FDG PET-CT has provided less effective than standard rituximab maintenance for advanced stage FL [73]. Therefore, negative ePET upon completion of induction therapy can be used to reduce toxicities created by consolidative or maintenance therapies for HL and DLBCL. While FDG PET-CT is superior to CT alone in post-induction prognosis of FL, it cannot be used, at this point, to guide further therapy.

Patients who show either residual disease or suboptimal treatment response by ePET usually have less favorable outcomes. These patients, even if subsequently achieving complete response to salvage treatment, are at higher risk of developing later relapse [74,75]. The prognostic value of ePET-CT has been found to be more accurate than iPET, especially in DLBCL [76]. In describing persistent or recurrent metabolically active tumor, it is critical to communicate whether the sites of metabolically active tumor are a recurrence of previously active sites of disease, or the appearance of new foci of tumor that were not present on previous studies. This has significant prognostic and therapeutic significance and may affect eligibility for clinical trials. It is also critical to clarify whether extra-nodal sites of disease are present. Therefore, when interpreting an FDG PET-CT scan, it is important to compare the ePET to all previous FDG PET-CT studies in which active disease was present, and not just the most recent prior exam.

#### 2.2.7. Surveillance Imaging by FDG PET-CT

The use of FDG PET-CT in surveillance of patients with lymphoma is not recommended, unless there is a clinical reason to believe that recurrence, or of transformation of a known low-grade lymphoma, has occurred. If strong clinical features support that relapse or transformation may have occurred, FDG PET-CT can be helpful. However, with HL and NHL, false positive findings and a low additional benefit over clinical assessment argue against the routine of FDG PET-CT in surveillance [24]. In FL, even with minimal residual disease (MDR) post-induction, there is no benefit of FDG PET-CT over clinical monitoring for subsequent progression [28].

### 2.3. Mature B-Cell Lymphomas (BCLs)

#### 2.3.1. Diffuse Large B-Cell Lymphoma (DLBCL)

Diffuse large B-cell lymphoma (DLBCL) is an aggressive neoplasm, representing the most common histologic subtype of NHL, accounting for 30% of all cases of lymphoma. The median age at presentation of 65 years. Patients usually present with rapidly enlarging nodal masses, mostly in the head and neck or abdomen. Most patients present in advanced stage, with approximately 40% showing extra-nodal sites of involvement. The gastrointestinal tract is the most common extra-nodal site of disease, with involvement of the stomach, ileum, large bowel, and esophagus in decreasing order of frequency. Imaging features include hypermetabolic nodular or ulcerative bowel masses, diffuse infiltrative masses or polyps. Diffuse bowel uptake on FDG PET-CT is less related to disease involvement, as most cases have had negative endoscopy. On the contrary, focal lesions are more suspicious for malignancy [77]. Pancreatic involvement can be seen as diffuse pancreatic infiltration, resembling pancreatitis or as a focal lesion, mimicking primary pancreatic cancer. CNS involved often presents as hypermetabolic periventricular masses. Additional common sites if CNS involvement include the corpus callosum, thalami, basal ganglia and ependymal surfaces [78]. However, the use of MRI is the standard of care if CNS disease in the brain or spinal cord is suspected.

DLBCL is often aggressive in behavior but tends to be sensitive to chemotherapy, allowing for curative intent treatment. There are three major subtypes of DLBCL that are genetically heterogeneous and arise from different and distinct genetic pathways [79]. These include germinal center B-cell-like (GCB) DLBCL, activated B-cell-like (ABC) DLBCL, and primary mediastinal lymphoma (PMBL). FDG PET-CT is essential in the initial staging of DLBCL. It has been shown to have a good negative predictive value in obviating the need for bone marrow biopsy in patients showing abnormal no osseous uptake on baseline staging. As noted above, a negative interim FDG PET-CT in DLBCL should not be used to downgrade therapy to a less toxic regimen, and evidence of progression on iPET or persistent/progressive disease on ePET should drive biopsy confirmation before a change in management is considered. However, analysis of a large recently reported clinical trial (GOYA), a negative ePET is predictive of PFS and OS [80].

Clinical surveillance of patients well beyond the end of treatment is mandatory, as 20–40% of patients with DLBCL will relapse, mostly in the first 2 years. CT imaging for surveillance can be performed but is not necessarily required in patients without symptoms of recurrence. The intervals between follow-up assessments are extended if disease remains in remission after 5 years [5,81]. Surveillance by FDG PET-CT is also not indicated unless there is a clinical suspicion of recurrence. FDG PET-CT can also suggest histologic transformation of indolent lymphoma into DLBCL, usually manifested by an increasing SUVmax, size of lymph nodes and/or the development of new lesions, despite good response of other sites of disease [3]. An example of DLBCL with initial good response but subsequent recurrence is shown in Figure 2.

#### 2.3.2. Follicular Lymphoma (FL)

Follicular lymphoma (FL) is the most common indolent B-cell tumor, and the 2nd most common NHL after DLBCL, accounting for 20% of all NHL cases [78]. The median age at presentation is 60 years. Patients are often asymptomatic and rarely present with B symptoms. Clinically, patients have diffuse lymphadenopathy with occasional extra-nodal sites of involvement.

The clinical course of FL is variable [82]. Some cases are asymptomatic and are discovered incidentally. These patients require no treatment as their disease can wax and wane without treatment. Asymptomatic FL can be managed by a “wait and see” strategy, by monitoring for worsening of symptoms or progressive disease. Other patients may progress without intervention. FDG PET-CT often shows hypermetabolic non-contiguous deep adenopathy or a mesenteric mass without obstructive manifestations (“sandwich sign”) [83]. Extra-nodal sites can be seen in the form of hypermetabolic diffuse or multifocal bone marrow disease or hepatosplenomegaly, with or without discrete lesions [82].

Upstaging of disease by FDG PET-CT is more commonly seen in FL than with other lymphoma subtypes, particularly when extent of disease is presumed limited by CT or MRI (Figure 3). It has been reported that in approximately 19% of patients with FL (95% CI 14–23%), initial stage of is altered by FDG PET-CT, regardless of grade of disease [84]. This has implications for altered management, particularly in patients with suspected early-stage disease. The use of FDG PET-CT is supported in evaluation for suspected relapsed disease as well as for possible transformation into more aggressive subtypes, with some limitations. For assessment of histologic transformation, and SUVmax of >10 is 81% sensitive for histological transformation [85]. However, there is significant overlap in degree of metabolic activity in FL from low to high grade [86]. As such, biopsy is required to confirm histologic transformation [85,86]. FDG PET-CT can help identify candidate target sites for tissue sampling. FDG PET-CT is superior to CT alone in post-induction assessment of FL [87]. However, there is insufficient evidence at this point to support that FDG PET-CT can be used post-induction to guide further therapy. An additional limitation of FDG PET-CT in FL is in the identification of bone marrow involvement when the disease is diffusely present within the marrow [28].

#### 2.3.3. Marginal Zone Lymphoma (MZL)

Marginal zone lymphoma (MZL) is an indolent BCL, accounting for approximately 10% of lymphoma cases and is the 3rd most-common NHL subtype. These tumors arise from normal small lymphocytic aggregates that are dispersed in the mucosal tissue in response to any inflammatory/immune response. There are three different clinicopathologic subtypes, including splenic MZL (SMZL), nodal MZL (NMZL), and extra-nodal mucosal-associated lymphoid tissue (MALT) lymphomas. Patients usually present with localized disease, although disseminated disease can occur. MALT lymphoma differs from most other B-cell NHLs in that it has a predilection for stomach, salivary, thyroid, and lacrimal glandular tissues as well as the orbits.

In MZL, uncontrolled B-cell proliferation occurs secondary to repeated local antigen stimulation as a sequala of infections or autoimmune inflammatory disorders. Well-reported associations include H-Pylori infection with gastric MALT, Hashimoto thyroiditis and Sjogren syndrome associated with thyroid and salivary MZL, and chronic sinusitis with sinonasal lymphoma. In some cases, eradication of the offending stimulus may be satisfactory for tumor regression and resolution.

Imaging findings on FDG PET/CT in MZL depend on the organ involved. Across the board, approximately 70% of MZL are FDG avid [81]. Splenic MZL often presents with hypermetabolic splenomegaly, with or without discrete focal masses [83]. For MZL, high lesion-to-liver and lesion-to-blood pool activity on ePET (but not on staging or iPET) have been shown to be predictive of PFS, but no OS [88,89].

Earlier studies suggest a limited role for FDG PET-CT in MALT lymphoma due to its characteristically low FDG avidity, although this is controversial. Gastric MALT lymphomas may appear as hypermetabolic polypoidal or infiltrative gastric masses but are more frequently low in activity. Interestingly, extra-gastric MALT is associated with higher uptake compared to gastric MALT. MALT lymphomas of the orbit manifests as hypermetabolic enhancing ocular masses with propensity to involve lacrimal glands. Pulmonary MALT lymphoma, include multifocal hypermetabolic pulmonary nodules/consultations or masses as well as perilymphatic and peribronchial soft tissue infiltrates. Compared to other NHLs, MALT lymphoma has an excellent prognosis with low rates of relapse. However, higher SUVs have been found to correlate more with tumor aggressiveness and higher relapse rate. An additional role for FDG PET-CT is to differentiate between disease localized to the stomach, orbit, or sinuses vs. disseminated disease (Figure 4). Localized disease can be managed by radiation monotherapy. Disseminated disease may require systemic therapy [5,81,90].

#### 2.3.4. Mantle Cell Lymphoma (MCL)

Mantle cell lymphoma (MCL) is a distinct aggressive subtype of BCL with a variable clinical course. MCL compromises 7% of all NHL cases. MCL usually affects patients at their 6th–7th decade of life. Multifactorial risk factors have been linked to MCL. The most unique is chronic inflammation and autoimmune disorders caused by antigen stimulation, resulting in immunologic dysregulation. Lately, the classification of MCL has been revised with introduction of two major subtypes, leukemic (20%) and classic (80%). These subtypes are included in the most recent WHO classification of hematologic malignancies. The leukemic subtype is usually seen in the elderly population, with a more indolent course. Imaging features include splenomegaly and bone marrow infiltration, with infrequent lymph node involvement. Management is usually conservative, and patients are followed by a watchful strategy. The classic subtype is unfortunately the most common subtype, presenting in younger patients with a more aggressive course requiring intervention. Patients usually present with diffuse adenopathy and extra-nodal sites of disease including bone marrow, gastrointestinal tract, kidneys and, less commonly, splenic involvement. FDG PET-CT is typically performed for any new patient with MCL, although this has not been conclusively proven as necessary. There is some evidence that FDG PET-CT may add value to baseline staging by detection of extra-nodal sites of disease as well as to guide biopsy [91]. Patients with MCL have a much higher relapse rate compared to DLBCL and FL (Figure 5) [5,81]. FDG PET-CT has also been shown to be of value, particularly when combined with biological and clinical indicators, in predicting PFS [92]. Ultimately, this may potentially provide opportunities to identify high risk patients appropriate for a more aggressive course of treatment.

#### 2.3.5. Lymphoplasmacytic Lymphoma/Waldenstrom Macroglobulinemia (LPL/WM)

Lymphoplasmacytic lymphoma (LPL) is an uncommon indolent B-cell neoplasm, sharing features between lymphomas and multiple myeloma (MM). It is characterized by overproduction of IgM antibody with less frequent secretion of other immunoglobulins, such as IgG. Waldenstrom macroglobulinemia (WM) is a term used mainly for LPL cases presenting with predominantly bone marrow involvement. Incidence of WM is 3–4 cases per million people in the United States with median of age of presentation in the mid 40’s [93].

Until recently, it was believed that all monoclonal gammopathy of unknown significance (MGUS) disorders predispose to frank myelomas. However, according to the 4th edition of WHO classification of lymphoid neoplasms and hematologic disorders, there are two different clinical entities according to the type of secreted immunoglobulin. Non-IgM MGUS is the most common disorder and is the real precursor to MM. IgM cases are the disorders that predispose to NHL and are more related to LPL/WM, than MM. LPL patients usually present with variable clinical features, including bone marrow involvement and extra-nodal sites of disease. A substantial number of patients, however, are asymptomatic at the time of diagnosis. Diagnosis of LPL only can be made after exclusion of other IgM secreting lymphomas [94].

Osseous involvement in WM, is usually seen in the form of diffuse osteopenia, bone marrow expansion and endosteal scalloping. Compared to MM, areas of intertrabecular osseous lucency are common with WM but frank lytic lesions are rarely seen [95]. With LPL, extra-osseous sites of involvement include splenomegaly, with or without focal lesions, and diffuse adenopathy within the axilla, retroperitoneum, and axilla. The role of FDG PET-CT in LPL and in distinguishing this entity from WM has received very little attention in the literature and research. Because activity on FDG PET is relatively modest in both osseous and non-osseous lesions, FDG PET-CT not typically recommended as an important diagnostic tool. However, it has been our anecdotal experience that, despite the relatively modest metabolic activity in LPL/WM, LPL has a relatively characteristic appearance on FDG PET-CT. Lymph node involvement with LPL is typically ill-defined, with extensive diffuse extracapsular extension (Figure 6). Peripheral perineural involvement with LPL is common, often infiltrating along intercostal nerves as well as along the lateral paraspinous sympathetic plexus and into the lateral vertebral recesses and sacral neuroforamina. Perineural involvement can occur due to deposition of high circulating IgM, causing cranial nerve deficit, brain parenchymal lesions [96]. Bing-Neel syndrome is an extremely rare CNS presentation of LPL, caused by deposition of circulating IgM in the CNS [97]. FDG PET-CT may therefore facilitate baseline staging, detecting extramedullary sites of disease, and in identifying candidate biopsy sites. There is also evidence that FDG PET-CT may be useful in monitoring response to disease and predict the transformation into more aggressive tumors, such as DLBCL, which is seen in 10% of cases of IgM LPL [94,95,98].

#### 2.3.6. Multiple Myeloma (MM)

Multiple myeloma (MM) is the second most common hematologic malignancy after lymphoma, accounting for roughly 10% of all hematologic malignancies. It is also the most common primary malignancy of the bone. MM arises from uncontrolled proliferation of plasma cells in the bone marrow, resulting in either focal or diffuse osteolytic lesions. Extramedullary soft tissue masses can also be seen, either originating as primary extra-osseous soft tissue lesions or from cortical breakthrough of osseous lesions.

Non-IgM monoclonal gammopathy of undetermined significance (MGUS) is believed to be the precursor for all myeloma cases. With time, MGUS usually progress into smoldering MM (SMM), which represents a transitional stage between MGUS and MM. The rate of progression from MGUS to MM is approximately 1% per year. SMM can slowly or rapidly progress to MM at a rate of 10% per year for the first 5 years. Once patients develop frank MM, they usually present with CRAB symptoms: hyper**C**alcemia, **R**enal failure **A**nd **B**one lesions [99,100].

There are three systems for staging of MM. The Revised International Staging System (R-ISS) pertains to prognostic risk, such as overall survival (OS) and relies upon non-imaging features, namely serum levels of β2M, albumin and LDL, as well as the presence of high-risk chromosomal features by FISH analysis [101]. The original Durie-Salmon classification system relies to a minimal degree on imaging features (1, 2–3, or >3 lytic lesions) as well as several laboratory features. The revised Durie-Salmon Plus staging system includes more discriminating and functional imaging findings, including MRI and FDG PET-CT [102]. Imaging findings included in the Durie-Salmon Plus system are a more stratified number of focal lytic lesions (>5 mm diameter), with modifiers based on the presence of extramedullary disease and serum creatinine level. The staging criteria for MM are compared in (Table 4). Most patients with MM are assigned both an R-ISS and a Durie-Salmon (including Plus) stage.

Bone marrow biopsy remains the gold standard for the diagnosis of MM although imaging plays an important role and can help with directed bone marrow biopsy. Conventional radiography (X-ray skeletal survey) is associated with high false negative rate of 30–70%, leading to a significant underestimation of presence and extent of disease compared to MRI and PET-CT [102]. Approximately 30–50% of osteoid needs to be destroyed before it can be appreciated on plain radiography. Plain radiography is also limited in detection of diffuse marrow involvement as well as extramedullary sites of disease. Common benign conditions, such as severe osteopenia and degenerative bone cysts can mimic MM and create sources of false positive findings by plain radiography. Nowadays, whole body MRI or CT and FDG PET-CT have essentially supplanted plain radiography in the imaging assessment of MM. FDG PET-CT offers a whole-body assessment with higher sensitivity compared to MRI in detecting extramedullary sites of disease [103]. Furthermore, MRI is more limited in detecting destructive lesions than FDG PET-CT, which has the added value of CT for ease in characterizing extend of lytic disease [104].

The primary limitation of FDG PET-CT in staging MM is a lower sensitivity than MRI in patients presenting with diffuse micronodular marrow infiltrative disease (Figure 7) [105]. However, diffuse increased activity associated with the marrow is not specific for MM, and can be confounded by other benign process, such as stress hematopoiesis from anemia or infection, treatment with granulocyte colony stimulating factor (GCSF) or other marrow stimulant drugs. As such, it can be difficult to determine whether increased activity is related to myeloma infiltration vs. marrow stimulation. Although marrow infiltration with MM is typically associated with higher FDG uptake compared to other non-malignant etiologies, bone marrow biopsy remains the only definitive answer in these cases [106,107,108].

In utilizing FDG PET-CT for staging and in assessment of level of progression along the spectrum from MGUS to frank MM, it is important to note and to report several factors that are critical in those assessments. This should ideally be provided in the form of a structured report with specific relevance for MM. These factors include the metabolic activity in background areas of red marrow, any focal hypermetabolic bone lesions suspicious for MM, the location and approximate number of lytic lesions > 5 mm diameter (defined as 1, 2–4, 5–20, or >20), and the presence of extramedullary sites of disease. MGUS and SMM are not usually associated with altered metabolic activity on FDG PET-CT. Any focal intramedullary or extramedullary lesion seen is highly predictive of transformation into active MM.

FDG PET-CT may have prognostic significance in portending more aggressive disease. The presence of more than three focal lesions and the presence of extramedullary sites of disease are usually associated with less favorable outcomes (Figure 8). Magnitude of metabolic activity on baseline FDG PET-CT has also been associated with poor outcomes, reported as >SUVmax of 7.1 on baseline scans, or >4 on post-treatment scans [109,110]. The degree of metabolic activity measured on PET-CT correlates with cellularity and plasma cell ratio concentration on biopsy, suggesting an estimate of disease burden [111].

FDG PET-CT can assess treatment response in the form of decreased metabolic response before structural changes occur. Lytic lesions, although showing resolution of metabolic activity on FDG PET-CT, often remain lytic indefinitely and do not heal with successful treatment, as opposed to lytic osseous metastatic disease, such as that with breast cancer, which becomes sclerotic with treatment. The reasons for this are not clear and cannot be explained by a loss of osteoblastic stem cells because fractures, even when involving an area of lytic disease, will undergo healing.

The Deauville scoring system was predominantly implemented for assessment of soft tissue sites of disease. However, it has been recently adopted by the International Myeloma Working Group (IMWG) for assessment of bone marrow and focal lesions (bone or extramedullary) for categories of complete metabolic response (CMR) and partial metabolic response (PMR) [112], as shown in Table 5.

In assessing response to treatment, FDG PET-CT may have an advantage over MRI. While conventional MRI findings of active disease (dark on T1, high signal intensity on STIR) may take years to resolve following successful treatment, resolution of metabolic activity on FDG PET-CT with successful treatment is relatively rapid [113]. However, in a recent systematic meta-analysis with direct comparison of MRI and FDG PET-CT for assessment of treatment response in the same subjects, the pooled sensitivity of whole-body MRI, at 87% (95% CI, 75–93%) was higher than for FDG PET-CT at 64% (95% CI, 47–68%), although the difference was not statistically significant (*p* = 0.29) [114]. However, the specificity was significantly lower (*p* < 0.001) for whole body MRI, at 57% (95% CI, 37–76%) than for FDG PET-CT, at 82% (95% CI, 75–88%). There was no significant difference in sensitivity or specificity of whole-body MRI without DWI when compared to that with DWI [114].

### 2.4. Hodgkin Lymphoma (HL)

Hodgkin lymphoma (HL) is a family of characteristic BCLs, accounting for less than 15% of all lymphomas in western regions. The WHO classifies HL into 2 major categories, classical HL and nodular lymphocyte predominant HL. Within classical HL, which accounts for 95% of all HL, subtypes include nodular sclerosis, mixed cellularity, lymphocyte rich and lymphocyte depleted. All subtypes of HL are hypermetabolic on FDG PET-CT. Some differences have been reported in magnitude of FDG uptake between the subcategories of HL [115]. However, significant overlap in metabolic activity precludes the use of FDG PET-CT in differentiating between the subtypes of HL. HL has two age-related peaks. The earliest peak presents at age of 20–30 and the other peak presents at age of 60. Although HL represents less than 1% of all cancers, it is considered one of the most curable cancers in the young population, with cure rate reaching more than 90% in classical HL [116]. HL is characterized by presence of Reed-Sternberg cells. Clinically, patients present with diffuse contiguous adenopathy in the neck, chest, and axillary region. Extra-nodal sites are less commonly seen with HL than with NHL. The most common extra-nodal site of disease in HL is the skeleton. Patients presenting with localized disease usually achieve remission in more than 90% of cases. A lower percent (60%) achieved remission when presenting with advanced or extra-nodal disease [5,81,106].

As detailed above, FDG PET-CT is essential in management of HL (Figure 9). Initial staging is performed according to the Lugano staging classification system (Table 2). FDG PET-CT leads to an upstaging in 10–40% of patients with HL and alters management in up to 20% [28,117,118]. For lower stage HL where radiation treatment is planned, FDG PET-CT alters the treatment field in a significant degree, compared to CT [119]. Interim PET (iPET) for HL utilizes the Lugano modification of the Deauville scoring system (Table 3). For HL, a negative iPET can justify reduction to a less toxic therapeutic regimen without affecting outcome [53,54,55]. Escalation to a more aggressive therapy in the face of a positive interim scan in high-risk patients with HD has been shown to improve outcome [57]. With iPET or end-of-treatment PET (ePET), many patients with HL have a residual fibrotic mass. FDG PET-CT can identify poor responders, with suboptimum drop in SUV according to Deauville criteria, and indicate that further treatment is appropriate (Figure 9).

### 2.5. T-Cell Lymphoma (TCL)

T-cell lymphoma (TLC) arises from T-cells/natural (NK)killer cells that originate in the bone marrow and undergo cellular mutation in thymic tissue. They represent 15% of all NHL cases. There are different classification strategies for T-cell lymphomas. These include: aggressive vs. indolent, peripheral vs. cutaneous, leukemic, nodal or extra-nodal, as well as classifications based on molecular markers, cellular receptors and cytokine secretions. Viral associations are common, including Ebstein-Barr virus (EBV), and human T lymphotropic virus-1 (HTLV1). The most common histologic subtype is peripheral T-cell lymphoma not otherwise specified, (PTCL-NOS), which does not fit, histologically, with any specific subtype of lymphoma. Other common subtypes of T-cell lymphomas include anaplastic large cell lymphoma (ALCL), primary systemic type, and extra-nodal NK-T cell lymphoma-nasal type. Mycosis fungoides and Sezary syndrome are subtypes of cutaneous T-cell lymphoma (CTCL). These cutaneous subtypes can elude diagnosis because they are clinically similar to other benign skin disorders. If not recognized; however, these disorders can progress rapidly and aggressively.

Generally, TCL is less studied and, accordingly, less-well understood compared to BCL. Most TCLs and BCLs are FDG-avid on PET-CT. Although the degree of SUV uptake has been correlating with degree of aggressiveness in most BCLs, this is not true in TCLs. The use of FDG PET-CT is still mandatory in staging and treatment response assessment. TCLs usually have a poorer prognosis compared to BCLs with a 5-year survival rate of less than 20% (Figure 10) [120,121,122].

### 2.6. Leukemia

Leukemia is the third most common hematologic malignancy after lymphoma and MM. It is the most commonly encountered cancer in children, accounting roughly for 40% of all malignancies. Leukemia arises from hemopoietic cells, infiltrating the marrow and other hemopoietic tissues resulting in either focal or diffuse marrow replacing process. It also has slight predilection for extraosseous soft tissues. Leukemias have been traditionally classified according to their clinical behavior and cellular maturation into 4 main subgroups: acute lymphoblastic leukemia (ALL), acute myeloblastic leukemia (AML), chronic lymphoblastic leukemia (CLL) and chronic myeloblastic leukemia (CML) [123]. However, newer separate categories of precursor leukemia/lymphomas have been added. The category of precursor lymphoma/leukemias includes both those of B-cell and T-cell origin. For those of B-cell origin, this includes two entities, B-cell acute lymphoblastic leukemia/lymphoblastic lymphoma not otherwise specified (NOS) and B-cell acute lymphoblastic leukemia/lymphoblastic lymphoma with recurrent genetic abnormalities. Precursor T-cell leukemia/lymphomas include T-cell acute lymphoblastic leukemia/lymphoblastic lymphoma and NK-cell acute lymphoblastic leukemia/lymphoblastic lymphoma. These precursor lymphomas are characterized by high numbers of circulating lymphoblasts in the marrow, blood and, in some cases, infiltration of solid organs. The use of FDG PET-CT in precursor lymphomas has not been extensively studied but some evidence supports that high SUV lesions at baseline predicts a more aggressive course, and a negative scan after the first cycle of chemotherapy portends a good prognosis [124].

#### 2.6.1. Acute Lymphoblastic Leukemia (ALL)

ALL is the most common leukemia in pediatric population, while CLL is the most common leukemia in elderly. Clinically, ALL presents with stigmata of bone marrow infiltration, resulting in severe pancytopenia. FDG PET-CT is not routinely used in management of leukemias in children because of preferential involvement of bone marrow and rarely present with soft tissue masses. Marrow biopsy remains the gold standard for management of these disorders. FDG PET-CT may nonetheless be a beneficial noninvasive tool for assessing pediatric patients as well as adult patients who present with nonspecific symptoms, including bone pain and fever [125]. The pattern on FDG PET-CT in most cases of ALL is metabolic activity in the medullary space of the red marrow containing portions of the skeleton. This finding on FDG PET-CT may be multi-focal or diffuse, and often involves expansion of hypermetabolic red marrow distally into the long bones. ALL may or may not involve the spleen. FDG PET-CT can also detect extra-medullary sites of disease known as granulocytic sarcoma (GS), or chloroma, which is commonly found in patients with AML or relapsed ALL. GS may occur in the head and neck, breasts, kidneys, testes, skin, and lymph nodes. The presence of extramedullary sites of disease in leukemia is associated with poorer outcome and refractory treatment and should be identified as early as possible [126].

#### 2.6.2. Chronic Lymphocytic Lymphoma/Small Lymphocytic Lymphoma (CLL/SLL)

CLL/SLL is an indolent low-grade hematologic malignancy, representing the most common type of leukemia in the western communities and elderly. CLL and SLL are in the same spectrum of disease, with CLL representing circulating cells and SLL representing nodes and solid organ/soft tissue involvement. Most patients are asymptomatic and discovered incidentally on routine blood work or checkup. On imaging, patients usually show mild hepato-splenomegaly and varying degrees of lymphadenopathy. Metabolic activity in lymph nodes of CLL on FDG PET-CT is typically low, with an SUVmax of <4. CT can safely replace PET-CT in the initial staging of these tumors. Up to 8% of cases of CLL can undergo histologic transformation, known as Richter transformation or Richter syndrome (RS), into more aggressive lymphoma, usually DLBCL. Patients with RS usually present with rapidly enlarging nodes and increasing size of hepatosplenomegaly. FDG PET-CT is useful in confirming suspicion for transformation (Figure 11). On FDG PET-CT associated SUVmax with Richter syndrome (RS) transformation is usually >5, although there is some overlap in metabolic activity between transformed and non-transformed sites of CLL. FDG PET-CT can also identify the most accessible metabolically active lesion for biopsy which will lead lower false negative results. Of note, populations of untransformed CLL and transformed lymphoma may co-exist in the same patient. With cytotoxic chemotherapy directed against the transformed tumor, little response is often seen in the sites of untransformed CLL (Figure 12). Prognosis with CLL transformation is poor and median survival is a few months after diagnosis [126,127].

## 3. Conclusions

In the field of hematologic malignancies, FDG PET-CT plays a critical role in staging as well as interim and end-of-treatment assessment of therapeutic response, as well as in suspected recurrence. It is important that the imaging professionals be aware of the classifications, typical and atypical imaging features and important clinical information that must be conveyed that pertains to the specific patient and type of hematologic malignancy. The degree to which PET-CT contributes to the overall clinical management of patients with hematologic malignancies is dependent on many important factors that require dedication, attention to detail, ongoing self-education and good communication between the providers and imaging professionals. An understanding of the normal biodistribution and physiologic parameters that may alter the distribution of FDG are needed to accurately interpret the scans. The maintenance of a knowledge of the literature is critical for ongoing professional improvement and relevant usefulness of imaging studies. This typically requires both a familiarity with the radiology and nuclear medicine literature, but also with specific publications relevant to hematologic malignancies. It is important that the imaging provider be cognizant of the fact that the classification of subtypes of lymphoma and other hematologic malignancies is an ever-evolving process. Ongoing clinical trials and consensus recommendations continuously alter the indications and quantitative parameters of relevance with respect to FDG PET-CT in hematologic malignancies. Finally, and perhaps most importantly, it is critical to have a thorough understanding of the patient’s history, co-morbidities and the specific questions being asked by the referring providers. If diligence is applied to ensure that all of these parameters are achieved and maintained, the radiologist or nuclear medicine practitioner becomes a true member of the interdisciplinary health care team. Conversely, the referring providers should understand the limitations and advantages of PET-CT, communicate well with the imaging professionals, and ask specific questions that will guide the imagers in providing the most relevant and accurate information possible.

## Figures and Tables

**Figure 1 cancers-14-05941-f001:**
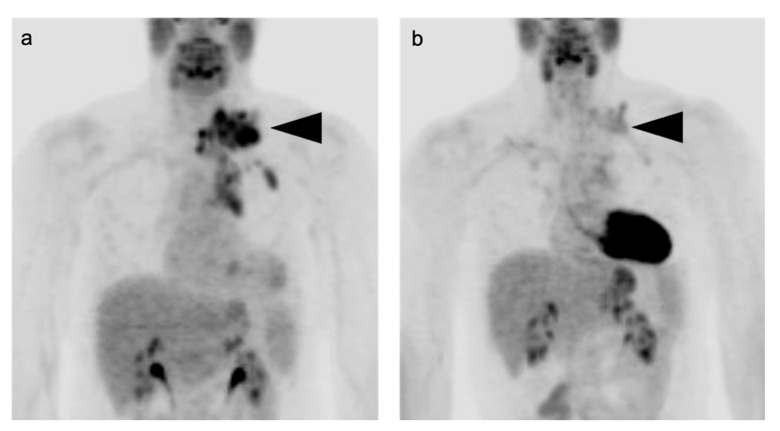
Rapid effect of chemotherapy on uptake of [^18^F] fluordeoxyglucose (FDG) in lymphoma. (**a**) A pre-treatment FDG positron emission tomography-computed tomography (PET-CT) maximum intensity project (MIP) image shows marked metabolic activity in a mass in the left lower neck, left mediastinum and left upper pulmonary lobe (black arrowhead); (**b**) 4 days following initiation of chemotherapy, a repeat FDG PET-CT MIP image shows a marked reduction in metabolic activity in the sites of tumor involvement (black arrowhead).

**Figure 2 cancers-14-05941-f002:**
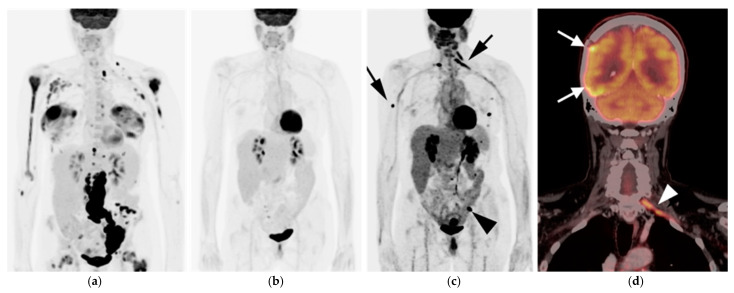
A 60-year-old female with diffuse large B-cell lymphoma (DLBCL). (**a**) At presentation, [^18^F] fluorodeoxyglucose positron emission tomography-computed tomography (FDG PET-CT) maximal intensity projection (MIP) image shows large hypermetabolic conglomerate abdominal nodal masses as well as left axillary nodes. Several extra-nodal sites are also seen, including bones and breasts; (**b**) On a MIP image after 6 cycles of chemotherapy, the same patient achieved complete metabolic response (CR); (**c**) MIP image after 2 years revealed relapsed disease with new lesions, including left external iliac node (black arrowhead), right proximal humerus (black arrow), linear uptake within the lower neck (black arrow); (**d**) Coronal fused PET-CT image, showed activity along left C7 nerve root (white arrowhead), consistent with perineural lymphomatosis which can be seen with aggressive lymphomas. White arrows show uptake in the right brain due to lymphoma involvement. Peripheral perineural spread increases the risk of central nervous system (CNS) involvement.

**Figure 3 cancers-14-05941-f003:**
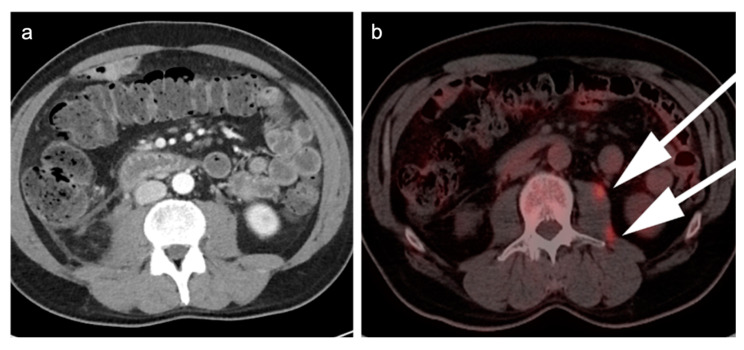
A 50-year-old-male with diagnosis of limited follicular lymphoma (FL) within an excised lymph node of the neck underwent baseline staging. (**a**) Axial contrast enhanced computed tomography (CT) image shows no disease in the abdomen.; (**b**) Axial fused [^18^F] fluorodeoxyglucose positron emission tomography-CT(FDG PET-CT) images, show two hypermetabolic sites of subsequently biopsy-proven FL in the left psoas muscle (white arrows). FDG PET-CT upstages disease in 19% of patients with FL, revealing occult sites not seen on other conventional imaging.

**Figure 4 cancers-14-05941-f004:**
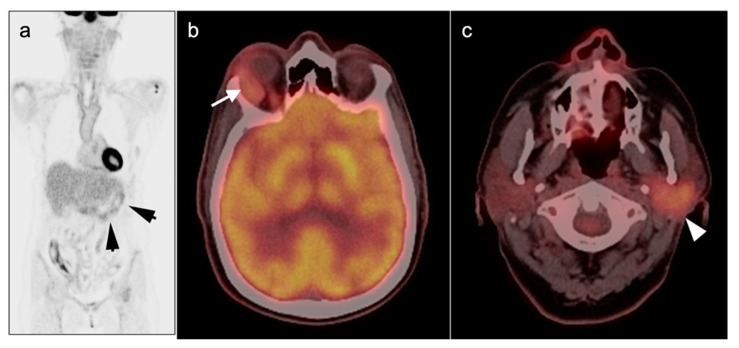
Three patients with mucosal associated lymphoid tissue (MALT) lymphoma. (**a**) [^18^F] fluorodeoxyglucose positron emission tomography-computed tomography (FDG PET-CT) maximum intensity projection (MIP) image shows minimal uptake in the stomach (black arrows) in a patient with biopsy-proven MALT lymphoma of the stomach. Gastric MALT lymphoma is often associated with low/mild uptake indistinguishable from normal physiologic gastric metabolic activity; (**b**) Fused FDG PET/CT axial images of the head in a patient with lacrimal MALT lymphoma (white arrow); (**c**) Fused FDG PET-CT axial images of the neck in an additional patient show a moderately FDG-avid left parotid MALT lymphoma (white arrowhead). Unlike gastric MALT, extra-gastric MALT lymphomas are often associated with higher levels of metabolic activity on FDG PET.

**Figure 5 cancers-14-05941-f005:**
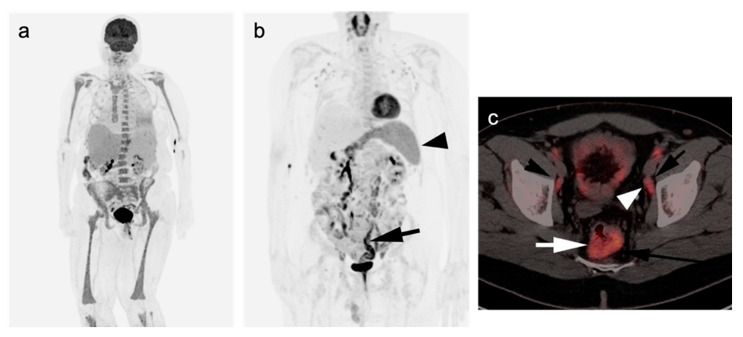
Two adult patients with mantle cell lymphoma. (**a**) In an older patient, [^18^F] fluorodeoxyglucose positron emission tomography-computed tomography (FDG PET-CT) MIP image shows mild uptake associated with the spleen and hypermetabolic and expanded distribution of red marrow compatible with mantle cell lymphoma, leukemic subtype; (**b**) In a younger patient with mantle cell lymphoma, FDG PET-CT MIP image shows diffuse adenopathy within neck, chest abdomen and pelvis, hypermetabolic splenomegaly (black arrowhead) and linear area of uptake within pelvis (black arrow) (**c**). Axial PET-CT in the same patient as (**b**) shows hypermetabolic lymphadenopathy (white arrowhead) and thickening and hypermetabolism within the distal sigmoid colon (white arrow). This was biopsy proven to be mantle cell lymphoma, classic subtype.

**Figure 6 cancers-14-05941-f006:**
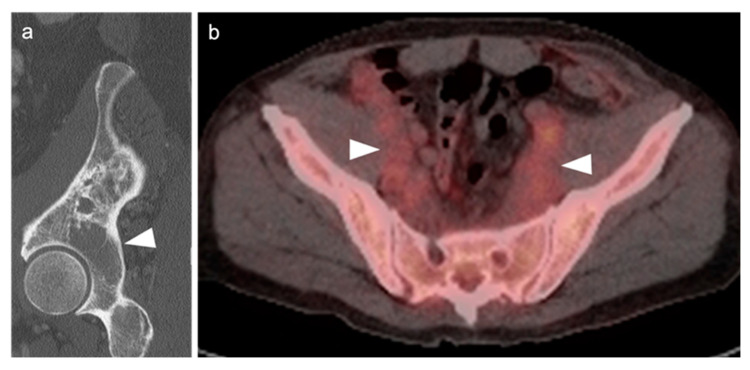
(**a**) A patient with Waldenstrom’s macroglobulinemia (WM) shows an expanded intertrabecular lucent marrow space (white arrowhead) on CT; (**b**) Axial [^18^F] fluorodeoxyglucose positron emission tomography-computed tomography (FDG PET-CT) image illustrating the typical appearance of lymphoplasmacytic lymphoma (LPL) nodal disease (white arrowheads) which is often ill-defined, infiltrative-appearing, and mild-moderately hypermetabolic.

**Figure 7 cancers-14-05941-f007:**
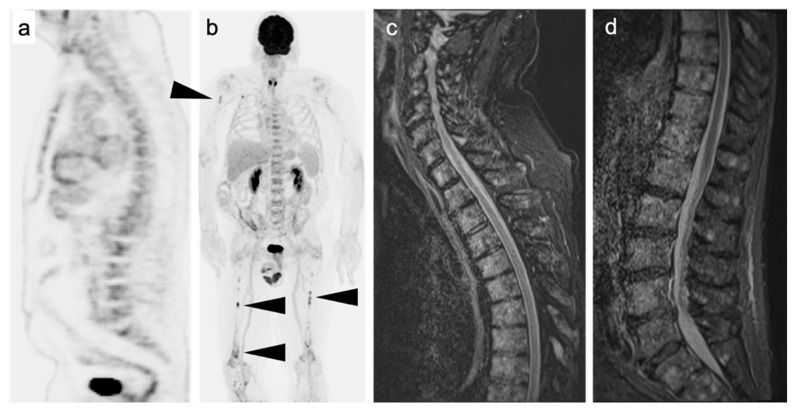
A patient with MM and micronodular disease. (**a**) [^18^F] fluorodeoxyglucose positron emission tomography (FDG PET) sagittal and (**b**) MIP images show non-specific moderate heterogeneous metabolic activity in the spine, but several focal lesions in the long bones (black arrowheads); (**c**,**d**) sagittal STIR weighted magnetic resonance imaging (MRI) of the thoracic and lumbar spine shows diffuse heterogeneous high signal intensity consistent with micronodular MM. FDG PET is less sensitive than MRI for detecting diffuse micronodular MM.

**Figure 8 cancers-14-05941-f008:**
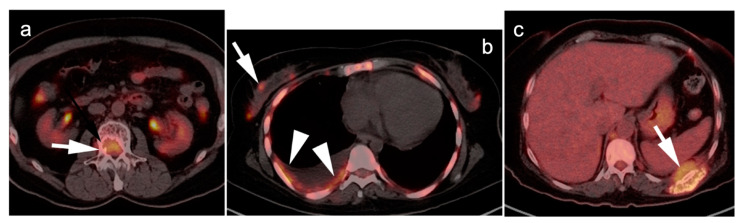
Aggressive myeloma with multiple extramedullary sites of tumor involvement on [^18^F] fluorodeoxyglucose positron emission tomography-computed tomography (FDG PET-CT). (**a**) A lytic lesion of a lumbar vertebra with an epidural soft tissue component (white arrow); (**b**) extensive subpleural tumor (white arrowheads) as well as multiple breast tumor implants (white arrow), and (**c**) a large lytic lesion of the rib with a soft tissue component (white arrow).

**Figure 9 cancers-14-05941-f009:**
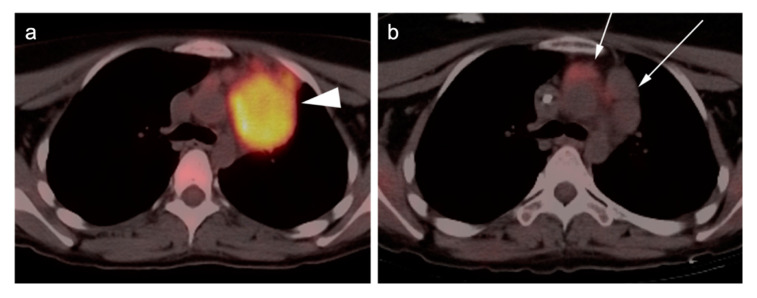
Hodgkin lymphoma (HL) before and after treatment. (**a**) An initial staging axial chest [^18^F] fluorodeoxyglucose positron emission tomography-computed tomography (FDG PET-CT) shows a hypermetabolic left anterior mediastinal mass in a young patient with HL (white arrowhead). (**b**) Post-treatment FDG PET-CT shows residual although smaller mass. Although metabolic activity has decrease with treatment (white arrows), there is a small focus of residual activity in a portion of the right anterior portion of the mediastinal mass concerning for persistent viable tumor. Activity within this region was significantly higher than that in the liver and the specific nodular hypermetabolic region had increased in size.

**Figure 10 cancers-14-05941-f010:**
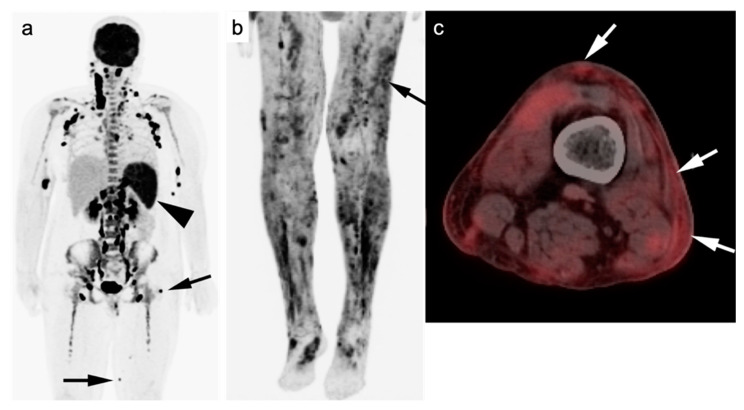
Two patients with peripheral T-cell lymphoma (PTCL) (**a**) An [^18^F] fluorodeoxyglucose positron emission tomography (FDG PET) maximum intensity projection (MIP) image of a patient with PTCL-NOS shows a typical pattern of multifocal disease involvement, including multiple nodal groups, the spleen (black arrowhead), and subcutaneous sites of involvement (black arrows); (**b**,**c**) Anaplastic large cell lymphoma (ALCL) demonstrates diffuse hypermetabolic dermal and subcutaneous lesions throughout the lower extremities shown on (**b**) FDG PET-CT MIP image (black arrow) and a (**c**) axial image of through the left thigh (white arrows).

**Figure 11 cancers-14-05941-f011:**
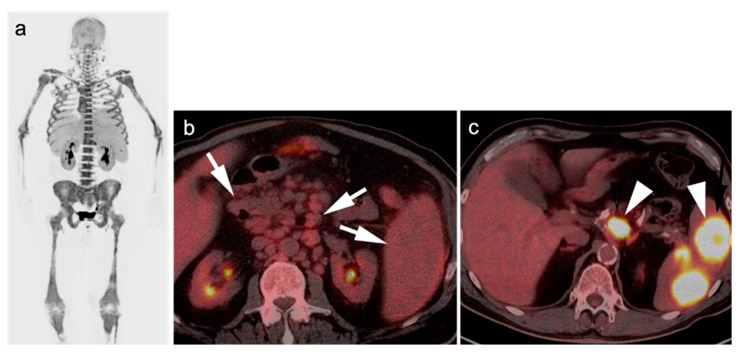
Three patients with leukemia. (**a**) Acute lymphoblastic leukemia (ALL): [^18^F] fluorodeoxyglucose positron emission tomography (FDG PET) maximum intensity project (MIP) image shows diffuse, intense uptake within the axial and appendicular skeleton, with marked expansion into the long bones, consistent with diffuse marrow involvement of leukemia. There is also splenomegaly. ALL rarely may rarely present with solid extraosseous masses or involved nodes. (**b**) Chronic lymphocytic leukemia (CLL): Fused axial FDG PET-CT axial image shows mild metabolic activity in an enlarged spleen and intraabdominal lymph nodes (white arrows); (**c**) Richter’s syndrome (RS) transformation in CLL. There is marked metabolic activity within a retroperitoneal node and multiple splenic lesions (white arrowheads). This was biopsy-proven RS with DLBCL subtype.

**Figure 12 cancers-14-05941-f012:**
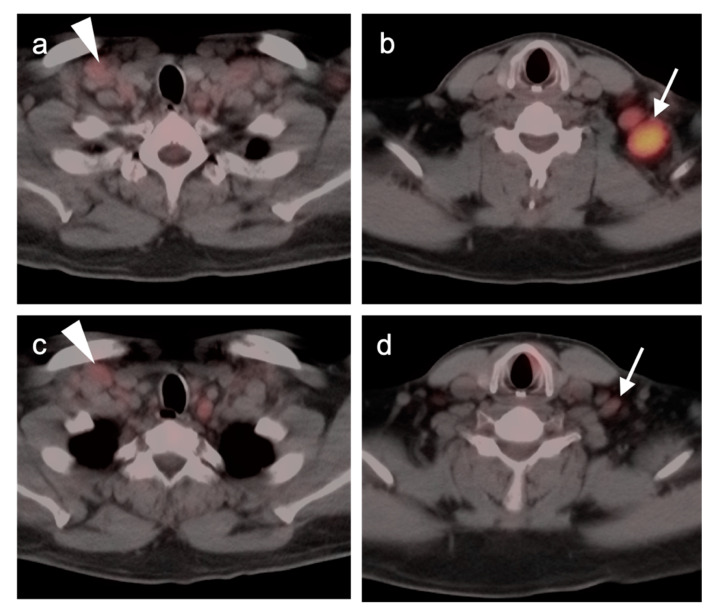
Coexisting CLL and Richter-transformed lymph nodes (DLBCL). Pre-treatment (**a**,**b**): (**a**) low-level metabolic activity in multiple bilateral supraclavicular lymph nodes (CLL, white arrowhead) as well as in (**b**) hypermetabolic transformed left level 5nodes (DLBCL, white arrow) in the same patient; Post-treatment (**c**,**d**): Following treatment for DLBCL, the CLL nodes have responded minimally ((**c**), white arrowhead) but the transformed lymph nodes show metabolic resolution and marked decrease in size ((**d**), white arrow).

**Table 1 cancers-14-05941-t001:** An overview of selected lymphoma subtypes encountered in clinical practice according to WHO classification of lymphoid neoplasms [1].

Precursor B-Cell Leukemia/Lymphoma		Precursor T-Cell Leukemia/Lymphoma		
B-cell acute lymphoblastic leukemia/lymphoblastic lymphoma not otherwise specified (NOS) B-cell acute lymphoblastic leukemia/lymphoblastic lymphoma with recurrent genetic abnormalities		T- cell acute lymphoblastic leukemia/lymphoblastic lymphoma NK-cell acute lymphoblastic leukemia/lymphoblastic lymphoma		
**Mature B-cell leukemia/lymphoma (non-Hodgkin lymphoma)**	**Mature T-cell/natural killer cell leukemia/lymphoma (Peripheral T-cell lymphoma**	**Hodgkin** **lymphoma (HL)**	**Post-transplantation lymphoproliferative disorder (PTLD)**	**Histocyte** **neoplasms**
Chronic lymphocytic leukemia/small lymphocytic lymphoma Monoclonal B-cell lymphocytosis B-cell prolymphocytic leukemia Splenic B-cell lymphoma/leukemia, unclassifiable Splenic diffuse red pulp small B-cell lymphoma Hairy cell leukemia Hairy cell leukemia variant Lymphoplasmacytic lymphoma Waldenstrom macroglobulinemia Monoclonal gammopathy of undetermined significance (MGUS), IgM and IgAPlasma cell myeloma Solitary plasmacytoma if the bone Extraosseous plasmacytoma Multiple myeloma Lymphomatoid granulomatosis Marginal zone lymphomas Nodal marginal zone lymphoma Extranodal marginal zone (MALT) lymphoma Splenic marginal zone lymphoma Pediatric marginal zone lymphoma Follicular lymphoma, in situ Pediatric follicular lymphoma Duodenal type follicular lymphoma Follicular lymphoma Large B-cell lymphoma with IRF4 rearrangement Mantle cell lymphoma Mantle cell lymphoma in situ Diffuse large B-cell lymphoma (DLBCL), germinal & activated B-cell types Primary DLBCL of the CNS Primary cutaneous DLBCL of leg DLBCL associated with chronic inflammation Burkitt lymphoma High grade B-cell lymphoma, not otherwise specified High grade B-cell lymphoma with MYC, BCL 2 and BCL6 rearrangement	Peripheral T-cell lymphomas Peripheral T-cell lymphoma, not otherwise specified (NOS) Angioimmunoblastic T-cell lymphoma Anaplastic large cell lymphomas ALK-positive ALK-negative Primary cutaneous large cell lymphoma Breast cancer (implant) associated anaplastic large cell lymphoma Cutaneous T-Cell Lymphomas Mycosis fungoides and variants Sezary syndrome Adult T-cell leukemia/lymphoma Primary cutaneous CD30+ lymphoproliferative disorders Subcutaneous panniculitis-like T-cell lymphoma Extranodal NK/T-cell lymphoma, nasal type Chronic active Ebstein Bar virus (EBV) infection T-cell prolymphocytic leukemia with cutaneous involvement Primary cutaneous peripheral T-cell lymphoma-NOS Primary cutaneous peripheral T-cell lymphomas (rare) ○ Primary cutaneous CD4+ small/medium T-cell lymphoproliferative disorder Primary cutaneous γ/δ T-cell lymphoma Primary cutaneous CD8+ aggressive epidermotropic cytotoxic T-cell lymphoma Primary cutaneous acral CD8+ T-cell lymphoma Primary intestinal T-cell lymphomas Enteropathy-associated T cell lymphoma Monomorphic epitheliotropic intestinal T-cell lymphoma Large granular leukemia-lymphoma of NK cells Lymphomatoid papulosis Systemic EBV+ T-cell lymphoma of childhood	Nodular lymphocyte predominant Classic Nodular sclerosis Lymphocyte rich Mixed cellularity Lymphocyte depleted	Plasmacytic hyperplasia Post-transplant lymphoproliferative disorders (PTLD) Infectious mononucleosis PTLD Polymorphic and Monomorphic PTLDs Classical Hodgkin lymphoma PTLD	Histiocytic sarcoma Langerhans histiocytosis Langerhans cell sarcoma Indeterminate dendritic cell tumor Interdigitating dendritic cell sarcoma Disseminated juvenile xanthogranuloma Follicular dendric cell sarcoma Fibroblastic reticular cell tumor Erdheim-Chester disease

**Table 2 cancers-14-05941-t002:** Lugano staging classification system for HL and NHL [8].

Stage	Site(s) of Disease Involvement
Limited stage disease	
I	Single lymphatic site
IE	Single extra lymphatic site without nodal involvement
II	Involvement of two or more lymphatic sites on same side of diaphragm
IIE	Extracapsular extension from a lymphatic site, +/− involvement of other nodes
II X (bulky)	Single node or nodal conglomerate >10 cm in any dimension or > than 1/3 the side-to-side diameter of the chest on CT
Advanced stage disease	
III	Involvement of nodal sites both above and below the diaphragm
IV	Diffuse or disseminated involvement with >1 extra-nodal site. Extra-lymphatic non-contiguous involvement with Stage II nodal disease. Any extra-lymphatic involvement with stage III nodal disease

**Table 3 cancers-14-05941-t003:** The Lugano modification of the Deauville Scoring System [4,8,51]. This classification system can be applied to metabolic activity of lesions (SUVmax) either for interim or end-of-treatment FDG PET-CT scans.

Deauville Category	Explanation of Metabolic Activity in Focal Lesions on FDG PET/CT	Treatment Response Interim FDG PET-CT	Treatment Response End-of-Treatment FDG PET/CT
1	No appreciable metabolic activity (for interim scan)	CR	CR
2	Mild metabolic activity ≤ blood pool	CR	CR
3	Mild metabolic activity > mediastinal blood pool and ≤ liver	CR	CR
4	Metabolic activity slightly to moderately higher than liver, no new lesions	PR if there is a significant reduction of metabolic uptake within previous sites of disease at interim PET-CT * SD if no change in metabolic activity ** PD if significant interval increases in metabolic activity compared to baseline or interim scan **	SD if no change in metabolic activity ** PD if significant interval increases in metabolic activity compared to baseline or interim scan **
5	Metabolic activity markedly higher than liver or new sites of disease	PR if there is a significant reduction of metabolic activity within previous sites of disease at interim PET-CT * SD if no change in metabolic activity ** PD if significant interval increases in metabolic activity compared to baseline or interim scan ** PD if new sites of disease **	SD if no change in metabolic activity ** PD if significant interval increases in metabolic activity compared to baseline or interim scan ** PD if new sites of disease **

CR (complete response); PR (partial response); SD (stable disease); PD (progressive disease). * Indicates chemotherapy sensitive disease. ** Indicates treatment failure.

**Table 4 cancers-14-05941-t004:** Comparison of Staging Systems for Multiple Myeloma.

Stage	R-ISS	Durie/Salmon	Durie/Salmon Plus	
I	Serum albumin > 3.5 g/dL Serum β2-microglobulin < 3.5 mg/L No high-risk cytogenetics Normal serum LDH	Hgb > 10.5 g/dL Serum Ca++ < 12 mL/dL No lytic lesions or single osseous plasmacytoma Low monoclonal-component production ○ IgG < 5 g/dL ○ IgA < 3 g/dL Urinary light chains < 4 g/24 h	MGUS IA (smoldering or indolent MM) IB	All negative No focal bone lesions or single plasmacytoma > 5 mm 2–4 focal lesions > 5 mm or mild diffuse spinal involvement
II	Neither Stage I or III	Neither Stage I or III and IIA—Serum Cr < 2 mg/dL Or IIB—Serum Cr > 2 mg/dl	IIA IIB	5–20 focal lesions > 5 mm or Moderate diffuse involvement Serum Cr < 2.0 mg/dL No extramedullary disease 5–20 focal lesions > 5 mm or Moderate diffuse involvement Serum Cr > 2.0 mg/dL Extramedullary disease present
III	Serum β2-microglobulin > 3.5 mg/L AND either ○ IIIA High risk cytogenetics or ○ IIIB Elevated serum LDH	Hgb < 8.5 g/dL Serum Ca++ > 12 mg/dL Imaging > 3 lytic lesions High monoclonal-component production ○ IgG < 5 g/dL ○ IgA < 3 g/dL Urinary light chains > 12 g/24 h	IIIA IIIB	>20 focal lesions > 5 mm Severe diffuse involvement Serum Cr < 2.0 mg/dL No extramedullary disease >20 focal lesions > 5 mm Severe diffuse involvement Serum Cr > 2.0 mg/dL Extramedullary disease present

**Table 5 cancers-14-05941-t005:** Refinement of end-of-treatment (ePET) criteria for multiple myeloma for bone marrow (BM) or focal lesions (FL, osseous or extramedullary) [112].

Response Definition	PET Response	Deauville Score
Complete metabolic response (CMR)	Uptake of BM or FL < liver	1–3
Partial metabolic response (PMR)	Persistent lesion with uptake > liver, but reduced in number or activity from baseline	4–5
Stable metabolic disease (SMD)	No change from baseline	
Progressive metabolic disease	New FL’s compared to baseline	

## Data Availability

There is no data reported.

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
