# Peer review of "PET-CT in Clinical Adult Oncology: I. Hematologic Malignancies"

_cancers, 2022, doi:10.3390/cancers14235941_

Round 1

Reviewer 1 Report (Previous Reviewer 2)

I am satisfied with the revisions performed,

thank You.

Author Response

Reviewer 1

Comments and Suggestions for Authors

( ) Extensive editing of English language and style required  
( ) Moderate English changes required  
(x) English language and style are fine/minor spell check required  
( ) I don't feel qualified to judge about the English language and style  

I am satisfied with the revisions performed,

Thank you.

Reviewer 2 Report (New Reviewer)

Overall, the review is very comprehensive and accurate in its description of the main indications  to the use of PET CT in Lymphomas.

However, I found some weaknesses in the evaluation of possible contraindications or side effects. The work should be supplemented by a paragraph on side effects or contraindications-limitations to the use of PET CT. In particular:

There is no mention of the radiation dose due to the combination of PET and total body CT. Considering the large number of young patients with lymphomas, more attention to the problem of radiation exposure should be placed.

In addition, there is no comment on the difficulties, complexity and high costs of reporting a hybrid imaging method that requires the integration of a radiologist with a nuclear physician for a complete and optimal evaluation. Please comment.

Finally, please add a note on the increasing use of total body MR with DWI, which is a radiation-free imaging modality  is extremely sensitive to lymphoproliferative tissues. It's role  is much underestimated. 

 Line 118. "MRI is usually typically reserved for assessment of the neural axis, 118 pelvic structures, and characterization of musculoskeletal involvement [5]."

This statement is not updated. MRI with DWI is increasingly used for the assessment of Lymphomas. Please mention two  recent papers:

MRI versus CT and PET/CT in the Preoperative Assessment of Hodgkin and Non-Hodgkin Lymphomas. Maccioni F., Calabrese A, Manganaro L et al. Hemato 20212(4), 635-644; https://doi.org/10.3390/hemato2040041

Whole-Body Magnetic Resonance Imaging: Current Role in Patients with Lymphoma Albano D,  Micci G, Patti C at al. Diagnostics 2021, 11, 1007. https://doi.org/10.3390/diagnostics11061007

Author Response

Reviewer 2

( ) Extensive editing of English language and style required  
( ) Moderate English changes required  
( ) English language and style are fine/minor spell check required  
(x) I don't feel qualified to judge about the English language and style  

  1. Overall, the review is very comprehensive and accurate in its description of the main indications to the use of PET CT in Lymphomas.

Thank you.

  1. The work should be supplemented by a paragraph on side effects or contraindications-limitations to the use of PET CT. In particular, there is no mention of the radiation dose due to the combination of PET and total body CT. Considering the large number of young patients with lymphomas, more attention to the problem of radiation exposure should be placed.

We have added a paragraph discussing the dosimetric implications of FDG PET-CT compared to total body CT. We have addressed this issue particularly with respect to young patients. We have added a discussion the advisability of a low dose CT in young patients to be performed with PET and have included a discussion of the possible advantages of FDG PET-MRI in dose reduction. We have also addressed the opportunities created by new generations of digital PET-CT scanners in enabling diagnostic scans performed with lower doses of radiopharmaceuticals.

  1. In addition, there is no comment on the difficulties, complexity and high costs of reporting a hybrid imaging method that requires the integration of a radiologist with a nuclear physician for a complete and optimal evaluation. Please comment.

This is a topic that we considered including but have deemphasized it because it is a politically charged issue, varies in its significance between different countries, and is relevant to “turf battles”. However, we agree that it should be included. In the US, radiologist are, as a requirement of their training, able to serve as responsible users for the receipt of radiopharmaceuticals used in PET. They are also trained in the interpretation of PET-CT. In the US, separate training programs for nuclear medicine are required to provide 6 months of training in CT interpretation and diplomats often elect to do additional fellowships in PET-CT. In the Netherlands, nuclear medicine training has also been integrated into diagnostic radiology. In the majority of other countries, radiologists are not certified to receive radiopharmaceuticals or to report the findings of PET-CT. However, imagers who are trained solely in nuclear medicine, and not in diagnostic radiology, face relative challenges in recognizing and clarifying the anatomic relevance of foci of uptake on PET-CT, in the identification of unsuspected but potentially clinically significant additional findings, and in being able to provide specific guidance on additional problem-solving imaging approaches. In these countries, a collaborative approach to interpreting PET scan is crucial even though this approach creates complexity and financial barriers in the optimal interpretation of these scans.

  1. Finally, please add a note on the increasing use of total body MR with DWI, which is a radiation-free imaging modality is extremely sensitive to lymphoproliferative tissues. It's role is much underestimated. We have expanded our discussion of the increasing use of total body MR with DWI in evaluating lymphoproliferative disorders. See additional relevance below.

  1. Line 118. "MRI is usually typically reserved for assessment of the neural axis, 118 pelvic structures, and characterization of musculoskeletal involvement [5]."

This statement is not updated. MRI with DWI is increasingly used for the assessment of Lymphomas. Please mention two recent papers:

MRI versus CT and PET/CT in the Preoperative Assessment of Hodgkin and Non-Hodgkin Lymphomas. Maccioni F., Calabrese A, Manganaro L et al. Hemato 2021, 2(4), 635-644; https://doi.org/10.3390/hemato2040041

Whole-Body Magnetic Resonance Imaging: Current Role in Patients with Lymphoma Albano D,  Micci G, Patti C at al. Diagnostics 2021, 11, 1007. https://doi.org/10.3390/diagnostics11061007

We have expanded the discussion to include these two references.

This manuscript is a resubmission of an earlier submission. The following is a list of the peer review reports and author responses from that submission.

Round 1

Reviewer 1 Report

These authors have reviewed the entire literature about the role of PET/CT in assessing hematological malignancies in adults and have provided an excellent and comprehensive write up on his topic. The authors have focused on various malignancies that are commonly diagnosed in the adult population with emphasis on lymphomas, myelomas and leukemias. The overall information provided by these authors is accurate and comprehensive and will be a good reference source for PET specialists about the state of the art in this discipline.

The only concern is the lack of information about optimal quantification of disease activity in this discipline of oncology. PET provides powerful quantitative data which is of great importance in such malignancies. Therefore, lack of any information about this important aspect of PET imaging is a limitation of the current version of the manuscript and should be emphasized in such reviews. In particular, the authors should make an effort in describing the differences between quantitative assessment of focal diseases that are seen in lymphomas versus diffuse bone marrow abnormalities in myelomas and leukemias. In particular, they should empathize the importance of regional versus global assessment in these varied malignancies. Adding this special segment will substantially enhance the impact of this critical review in the day-to-day practice of clinical oncology.

Reviewer 2 Report

This is a nice review article, part of a collection of reviews.

I have several improvements to suggest:

  • Some parts and concepts seem to be not updated with novel/recent imaging advancements. E.g. You stated that conventional radiography is still the reference standard for the staging of multiple myeloma. I suggest You to change this statement (MRI and/or PET-CT are the current reference standard for myeloma).
  • Regarding the comparison between MRI and PET-CT in Myeloma your statement is incomplete and partially not correct. MRI achieves better results than PET-CT in micronodular involvement (so-called 'salt'n'pepper' pattern) and in diffuse mild and moderate bone marrow involvement - PMID: 22921683). On the contrary PET-CT is better and faster in normalization after good response to therapies (PMID: 22921683). Implement and correct this section.
  • Moreover, about 10% of myeloma patients are FDG negative. Write something in this regard as a limitation of this tool.
  • Some parts of the paper are weel deepened and others are on the contrary lacking. Particularly, I feel that the section of Multiple Myeloma is not well deepened. Please implement it. PET-CT has nowadays a great role in staging, treatment response assessment and recurrence detection of the disease.
  • The image of Spinal MRI (T2w fat sat) of Myeloma is not so good and some artifacts are present. Moreover T1+C is a better tool or DWI whole body. Do you have another better sequence to substitute this?
  • The bibliographic literature references are poor for a comprehensive review such as this. Please implement it. (E.g. Suggested PMID: 15653595 a nice review of PET-CT in the treatment of Lymphoma. PMID: 33573573 and PMID: 30587527 two nice comprehensive review of imaging in myeloma... Find more to better prove scientifically your comprehensive review article).
  • In the simple summary please remove ";" and substitute with comma, the repetitive use in that context is inappropriate.

Round 2

Reviewer 2 Report

Dear Authors,

I am satisfied with the revisions performed.

Thank You